# Alternative Samples for Porcine Reproductive and Respiratory Syndrome Surveillance in an Endemic PRRSV-1-Infected Breeding Herd: A Descriptive Study

**DOI:** 10.3390/vetsci10090558

**Published:** 2023-09-05

**Authors:** Arnaud Lebret, Valérie Normand, Pauline Berton, Théo Nicolazo, Charlotte Teixeira Costa, Céline Chevance, Mathieu Brissonnier, Gwenaël Boulbria

**Affiliations:** 1Porc.Spective Swine Vet Practice, ZA de Gohélève, 56920 Noyal-Pontivy, France; 2Rezoolution Pig Consulting Services, ZA de Gohélève, 56920 Noyal-Pontivy, France

**Keywords:** swine, PRRSV, PCR, surveillance, serum, oral fluid, udder wipes

## Abstract

**Simple Summary:**

The aim of this study was to describe the rate of detection of PRRSV-1 by PCR in due-to-wean litters in an endemic infected herd using three different sample types (blood samples, family oral fluid and udder wipes). Rates of detection were compared after testing samples individually and after pooling. Blood samples gave the higher rate of detection even after pooling by five, confirming that, at this time, it seems to be the best sampling procedure.

**Abstract:**

Knowing porcine reproductive and respiratory syndrome (PRRS) status is essential for designing herd management protocols. For this, weaning-age pigs are a key subpopulation. Recently, different alternatives to blood sampling have been introduced because they are easier, welfare-friendly and cost-saving tools. Moreover, most of them allow the testing of more animals and seem to be more sensitive in low-prevalence scenarios. However, these studies were implemented mainly in PRRSV-2-infected herds. The first objective of our study was to compare the rate of detection of PRRSV-1 by RT-qPCR in individual serum samples, family oral fluid samples (FOF) and udder wipes (UW) collected the day before weaning. The second objective was to evaluate the suitability of pooling. The study was performed on a 210-sow farrow-to-finish farm which was PRRSV-1 infected and unstable. A total of 119 litters were sampled. The rate of detection of PRRSV-1 in blood samples, FOF and UW was 10.9%, 7.6% and 0.8%, respectively. The agreement between sera and FOF was almost perfect even if the detection capacity of sera was numerically superior to FOF. The Ct values of positive sera were statistically lower than those of FOF. Two modalities of pooling (1:3 and 1:5) were tested for sera and FOF. For sera, both modalities did not impact the PRRSV-1 status either at the litter level or at the batch one. On the other hand, whatever the modality (pooled by 3 or 5), most of the pools of FOF gave negative results, misclassifying many litters and batches.

## 1. Introduction

Porcine reproductive and respiratory syndrome virus (PRRSV) is one of the most economically important diseases in the swine industry and affects pig herds in many intensive pig production areas worldwide [1,2,3]. Its dramatic economic impact is caused by reproductive failures in sow herds and an increase in secondary infections and mortalities, as well as a decrease in growth performance in finishing units [4]. PRRS is caused by an RNA virus distinguished into two genotypes, namely, PRRSV-1 and PRRSV-2. It has to be mentioned that PRRSV-1 is the predominant species in Europe [5]. Before implementing any management strategy against the disease, knowing the status of the breeding herd remains an essential prerequisite. For a long time, sampling the blood of due-to-wean piglets was the recommended method to determine the stability of a sow herd [6]. More recently, many studies have investigated the relative significance of other diagnostic sample types. There were two main goals in the development of these alternative samples. The first one aimed to be more respectful of welfare as blood sampling is intrusive and therefore stressful. The second objective was to sample more animals without increasing analysis total cost, especially in low-prevalence scenarios.

In particular, individual or collective oral fluids of weaners [7,8], family oral fluids (FOF) [9,10], udder wipes (UW) [11], processing fluids (PF) [12,13], umbilical cord blood (UC) [14] and tongue tips (TT) [15,16] have been studied. Recently, the American Association of Swine Veterinarians (AASV) reviewed its classification, introducing PF and FOF as alternative options to blood, alone or in combination with sera [17]. Different scenarios are distinguished in the new classification, taking into account the expected status and the within-herd prevalence. For example, in the case of expected instability, sampling 30 piglets at weaning in at least four batches over a 90-day period remains the recommended protocol. When a low-prevalence scenario is considered, it is recommended to test six pools of 10 serum samples from 60 weaning-age pigs by RT-qPCR monthly for four consecutive months to classify breeding herds.

Most of the experiments aiming to evaluate the significance of new methods of sampling have been conducted in North America, targeting PRRSV-2. Few studies have been implemented in Europe in PRRSV-1-infected herds investigating the value of oral fluids (OF) [8,18], TT [15] and UC [14].

The aim of our study was first to compare the rate of detection of PRRSV in serum, FOF and UW in a PRRSV-1-infected and unstable herd. Secondly, we also evaluated the impact of pooling on detection capacity with each sample type.

## 2. Materials and Methods

### 2.1. Study Design

This descriptive study was conducted on a commercial 210-sow farrow-to-finish pig herd located in Brittany, France. The management of the farm was based on seven batches of around 30 to 35 sows each, and the age at weaning was 28 days on average. Replacement gilts were bought externally from a French PRRSV-negative nucleus herd.

This herd was confirmed PRRSV-1-positive unstable category I-A (according to AASV classification) before the beginning of the study and did not use any vaccination against PRRSV in sows or in piglets due to the farmer’s own decision.

Four consecutive batches were included, and, in each of them, 30 litters were sampled. The sampling was performed for regular PRRSV-1 monitoring by the veterinarians in charge of the follow up on the sanitary status of the farm.

### 2.2. Sample Collection

The samples were collected between August and November 2021. 

Within each batch, the day prior to weaning, in the morning, 30 litters were sampled using the following methods:-Blood from one piglet per litter, targeting the weakest piglet within the litter 

This sampling procedure for bleeding was performed in accordance with AASV recommendations [17] as the status of the farm was already known.

Blood samples were collected from the cranial vena cava in plain test tubes using one sterile needle per piglet. Samples were kept in cool storage (4 °C to 8 °C) until submission to the lab;

-FOF

FOF were collected by presenting an untreated cotton rope to the sow and its piglets, without training, the day before sampling. One end of the 50 cm rope (0.8 cm diameter) was knotted and attached with pliers to the farrowing crate, near the sow’s head. The other end ended at the shoulder level of the smallest piglet of the litter. After 30 min presentation, the wet portion of the rope was inserted into a plastic bag and manually wrung to collect sow and piglets’ oral fluid. After that, the corner of the bag was cut, and the oral fluids were transferred into a 10 mL tube and kept in cool storage (4 °C to 8 °C) until submission to the lab;

-UW

UW were collected by wiping all the underline skin of the sow’s udder with a 50 cm gauze (untreated and sterile cotton) previously impregnated with 5 mL of phosphate-buffered saline (PBS). The objective of this sample type was to indirectly collect piglets’ oral fluid after suckling. After collection, gauzes were inserted into a plastic bag and kept in cool storage (4 °C to 8 °C) until submission.

All samples were submitted to the laboratory within three hours on the day of sampling. 

### 2.3. Diagnostic Testing

Diagnostic tests were performed at Labofarm (Finalab Veterinary Laboratories Group, Loudéac, France). All samples were analyzed individually. Then, pools (1:3 and 1:5) were also analyzed. Only blood and FOF samples were tested for pooling.

Blood samples were centrifuged to separate serum (4500× *g* for 5 min) at room temperature. Two hundred µL of the supernatant was used for RNA extraction.

One mL of FOF was centrifuged for 10 min at 95× *g* at room temperature for sedimentation of big particles. Two hundred µL of the supernatant was used for RNA extraction. The content of the UW was suspended by kneading the wipes in 50 mL of PBS for approximatively 10 s. Two hundred µL of the suspension was then used for RNA preparation.

RNA was extracted using an Indimag Pathogen Kit (Indical Bioscience, Leipzig, Germany) following the manufacturer’s instructions.

All samples were tested for PRRSV RNA using an ADIAVET PRRSV REAL TIME kit (BioX Diagnostics, Rochefort, Belgium) in just one batch. A sample was considered positive if the cycle threshold (Ct) value was ≤ 40 and the curve had a specific exponential look.

### 2.4. Pooling

In each batch, PRRSV-1-negative samples in sera and FOF were combined and vortexed to form a homogenous negative sample for each sample type. Then, pools of 1:3 and 1:5 were created by diluting one part of all positive samples (sera and FOF) with, respectively, two or four parts of the PCR-negative homogenate. 

### 2.5. Data Analysis

#### 2.5.1. Comparison of the Rate of Detection between Sample Types and Agreement between Them

The agreement between sera and FOF was assessed at the litter level using a concordance test (kappa statistics) using publicly available software (https://idostatistics.com/cohen-kappa-free-calculator accessed on 17 November 2021).

#### 2.5.2. Evaluation of Pooling Ability to Detect PRRSV-1

At the batch level, the ability of pools (1:3 and 1:5) to detect the virus was assessed. The relation between the individual’s Ct value and pool’s Ct value was assessed using linear models. A *p*-value < 0.05 was considered as significant. Then, the Spearman coefficient between the individual Ct value and pooled Ct value was determined. For each statistical analysis, the different levels of pooling (1:3 and 1:5) were taken into account. All analyses were realized using RStudio (v.2023.06.0).

## 3. Results

### 3.1. Tests Abilities to Detect PRRSV

In total, 120 litters were sampled in four batches. In one litter in batch 2, we could not collect enough oral fluid. That was the reason why only 119 samples of serum, FOF and UW were compared.

In each batch, at least one sample type was positive, confirming the unstable status of the farm (Table 1). 

It was possible to detect a minimum of one positive sample per batch with sera and FOF but not with UW, with only one positive sample out of 119 in total. This means that UW were unable to detect PRRSV in three out of four batches (UW results are not presented anymore in the rest of this paper due to their poor capacity to detect PRRSV in the conditions of this trial).

In total, we found, respectively, 13 and 9 positive samples out of 119 in sera and FOF. Sixteen different litters tested positive with at least one sample type (Table 2). The value of the Cohen’s kappa between the sera and FOF was 0.84, indicating an almost perfect concordance between both sampling types (Table 2).

### 3.2. Comparative Ct in Samples Analyzed Individually

Ct values for blood samples and FOF are presented in Figure 1. 

The lowest Ct values in blood and FOF were 24 and 31, respectively. Ct values were significantly lower in sera than in FOF (*p* = 0.0006).

### 3.3. Evaluation of Pooling

Pooling was evaluated only for sera and FOF due to the lack of positivity with UW. It was possible to test pooling with all but one positive serum (due to insufficient quantity) and with all positive FOF. In Table 3, Ct values of individual and pooled samples are listed.

Statistical analyses were only performed for sera due to the small number of positive FOF. For sera, there was a strong correlation between individual Ct and pool Ct values (r = 0.96, *p* < 0.001). Indeed, individual Ct values significantly impacted the results after pooling (*p* = 0.007). Two out of 12 serum samples returned negative after pooling by 3 or 5. This result did not impact the qualification of the batch. Regarding FOF, after pooling by 3, seven out of nine samples returned negative, misclassifying two batches out of four. After pooling by 5, eight out of nine samples returned negative, misclassifying three batches.

## 4. Discussion

Monitoring PRRSV on farms allows determination of the herd status regarding shedding and exposure. For this, sera still remain the reference sample [17]. RT-qPCR is the most commonly used test for the diagnosis of PRRSV because of its high sensitivity and specificity. In the conditions of our study (only one herd and four batches selected), sampling serum from one piglet per litter or one family oral fluids sample per litter was sufficient to classify this production herd as unstable in a high-prevalence scenario according to AASV recommendations [6,17]. In this case herd, instability and high prevalence was previously demonstrated with persistent shedding and PRRSV-1-positive suckling piglet detection in all controlled batches before inclusion.

Sampling FOF is a way to increase the number of pigs sampled, and their collection is easy, quick and not stressful. Using FOF for detection of PRRSV has been well documented in the United States (US) [10,19]. It is currently a part of the AASV diagnostic guidelines for classification of breeding herds regarding PRRSV status [17]. Our results confirmed the significance of FOF compared to serum at the litter level and at the batch level. The Cohen’s kappa calculated was almost the same as the one calculated in a previous study conducted in PRRSV-2-infected farms [19]. However, the conditions of both studies were different, especially regarding sample collection. Indeed, Almeida et al. [19] sampled all piglets in each litter, FOF were collected and the results were compared between both sampling methods at the litter level. In our study, we sampled serum from only one piglet per litter, and we could have missed some positive litters.

In a previous study, our team already demonstrated that piglets’ collective oral fluid (cOF) is an interesting alternative for detecting PRRSV-1 in unstable herds at the litter and at the batch level [8]. However, the success rate of sample collection depends on several factors, particularly the age at weaning and previous training of the piglets [20]. In a previous study, Almeida et al. [19] also showed that collecting FOF was easier than collecting cOF. They assumed that piglets have a tendency to mimic the behavior of their mother, who interacts first with the rope. In our study, we also had a good success rate of collection of FOF (119/120), but we did not compare it with that of cOF.

Despite this fact, it has to be noticed that Ct values were significantly higher in oral fluids than in serum samples, as was previously reported in other studies [8,10,21]. The lowest Ct value in our FOF positive samples was 31.4, which makes the probability of being able to sequence the virus isolated from oral fluid samples very low [8].

Regarding UW, we found a very poor capacity of this sample type to detect PRRSV. This is in agreement with previous studies, which had a poor correlation between sera, PF and UW in detecting PRRSV-2 [22]. It is also the reason why the AASV did not retain this sample type in the new classification [17]. Another explanation could be that, in our study, we used 50 mL of PBS as support media to wash the gauze before kneading to collect fluids. It is possible that, with this methodology, we diluted the number of viral particles and decreased the sensitivity.

Finally, the effects of pooling serum and FOF samples on PRRSV detection were evaluated. In order to test a large number of pigs, pooled samples are routinely used for the monitoring of PRRS, especially with the aim of lowering the cost of analysis [8,10,23,24,25]. The significance of pooling serum samples has been widely investigated, and there is no longer any debate at this time. In our study, we chose to test two modalities of pooling (1:3 and 1:5) according to previous studies [10,21]. We did not test other modalities as other researchers did (1:10, for example). Regarding FOF, our results are in contradiction with the study of Osemeke et al. [10]. Indeed, they showed that, for instance, in a low-prevalence scenario, pooling FOF up to 1:10 was valuable. Even though they demonstrated an increase in Ct values after pooling, it did not change the final classification of the farm [10]. In our study, pooling FOF led to the misclassification of three batches out of four, which is not acceptable. We can assume that the discrepancies between both studies were due to the differences in the Ct values of our samples.

## 5. Conclusions

In the conditions of our study, conducted in one specific farm infected with a particular PRRSV-1 strain, FOF seem to be a good alternative to blood samples, but only when analyzed individually and not after pooling. All UW but one returned negative, showing that this kind of sample is not suitable for PRRSV-1 surveillance. Finally, we confirmed that blood samples give the higher rate of detection even after pooling by five, confirming that, at this time, it seems to be the best sampling procedure. Further investigations in herds with different PRRSV prevalence rates are needed to support our findings.

## Figures and Tables

**Figure 1 vetsci-10-00558-f001:**
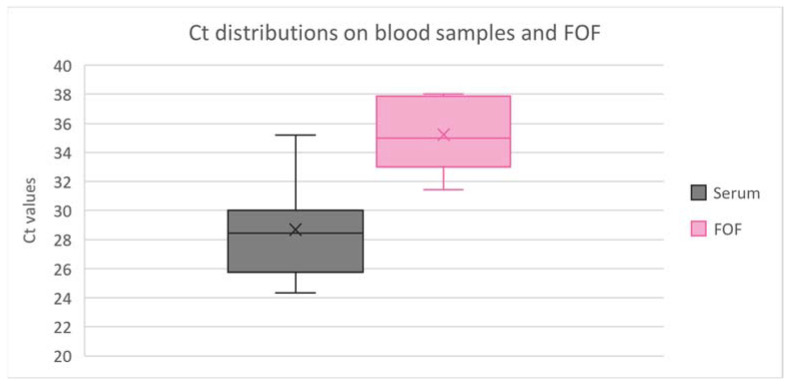
Distribution of cycle threshold (Ct) values for detection of PRRSV-1 from positive blood samples and positive family oral fluids (FOF) samples using RT-qPCR. Boxplots show median, quartiles, minimum and maximum values.

**Table 1 vetsci-10-00558-t001:** PCR results of the three sample types at the batch and at the litter level.

	No. of Litters	RT-qPCR +		
		Serum	FOF	UW
Batch 1	30	3	1	0
Batch 2	29	7	5	0
Batch 3	30	2	2	1
Batch 4	30	1	1	0
	119	13	9	1

**Table 2 vetsci-10-00558-t002:** Comparison of PRRSV-1 RT-qPCR detection in serum and FOF from litters of due-to-wean piglets.

		Serum		
		NEG	POS	Total
FOF	NEG	103	7	110
	POS	3	6	9
	Total	106	13	119

**Table 3 vetsci-10-00558-t003:** Ct values of sera and FOF samples (with batch identification) tested individually, pooled by 3 and pooled by 5 (ND = not done).

SERA					FOF				
Sample Identification	Batch	Ct Individual	Ct Pool 1:3	Ct Pool 1:5	Sample Identification	Batch	Ct Individual	Ct Pool 1:3	Ct Pool 1:5
Serum-1	1	24.9	28	28.9	FOF-1	1	31.4	32.5	34.8
Serum-2	1	24.3	28	28.9	FOF-2	2	33	>40	>40
Serum-3	1	25.5	28.3	29.4	FOF-3	2	38	>40	>40
Serum-4	2	26	28.4	29.2	FOF-4	2	38	>40	>40
Serum-5	2	33	34.4	37	FOF-5	2	35	35.4	>40
Serum-6	2	28	31.2	32	FOF-6	2	33	>40	>40
Serum-7	2	30	33.8	34.9	FOF-7	3	36.8	>40	>40
Serum-8	2	30	32.8	34.3	FOF-8	3	37.8	>40	>40
Serum-9	2	30	>40	>40	FOF-9	4	34.2	>40	>40
Serum-10	2	29	ND	ND					
Serum-11	3	35.2	>40	>40					
Serum-12	3	28.4	30.7	31.6					

## Data Availability

Not applicable.

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
