# Peer review of "Alternative Samples for Porcine Reproductive and Respiratory Syndrome Surveillance in an Endemic PRRSV-1-Infected Breeding Herd: A Descriptive Study"

_vetsci, 2023, doi:10.3390/vetsci10090558_

Round 1
Reviewer 1 Report
This manuscript is to study the detection rate of PRRSV-1 using three different sample types (serum samples, FOF, and UW) in pig farms known to be affected by PRRSV-1. The detection rates were compared between individual testing and pooled testing of the samples. The study found that serum samples had the highest detection ability in individual analysis, while FOF also showed good detection ability. However, the detection ability of UW was poor and could not reliably detect PRRSV. Additionally, the study evaluated pooled analysis of the samples and found some differences between individual and pooled analysis. Serum samples still had a high detection rate after pooling, while the detection ability of FOF was lower after pooling. This manuscript provides useful information on PRRSV detection, but further research is needed to validate and improve the results. Improvements in the interpretation and discussion of the results, as well as more experimental details, will enhance the quality and comprehensibility of the manuscript.
However, there are still some issues that need to be addressed in this research, including:
1 In the abstract, the author should provide pertinent information to elucidate why alternative testing methods exhibit higher sensitivity during periods of low epidemic prevalence, why sample pooling is deemed necessary, and explicate the influence of pooling on the outcomes.
2 In the introduction, it would be appropriate for the author to provide an introduction to the epidemiological characteristics of PRRSV, including its significant impact on pig herd health and productivity, as well as its global prevalence. The author can also mention the two subtypes of PRRSV, namely PRRSV-1 and PRRSV-2, and elaborate on their differences and implications.
3 In the "2.1. Study design" section of the Materials and Methods, please explain why this specific pig farm was chosen for the study, including background information on PRRSV-1 infection and instability at the site, as well as the reasons for not using PRRSV vaccines at the farm.
4 In the "2.4. Pooling" section of the Materials and Methods, explain why the pooling ratios of 1:3 and 1:5 were chosen, and discuss the impact of these ratios on virus detection capability.
5 Line 153: The result indicates that the Ct values for the blood and FOF samples are 24 and 31, respectively. However, the significance of these values in PRRSV detection and an explanation for them are not provided. It would be helpful to have more information about Ct values to gain a better understanding of the result.
6 Lines 213-219: This section discusses the advantages and limitations of using FOF for PRRSV monitoring, as well as the effectiveness of using pooled samples. However, there is not enough explanation and discussion provided to support these conclusions, and no specific information or data is provided regarding these impacts.
7 In the conclusions, it is noted that the research results were obtained under specific conditions. However, there is a lack of sufficient information provided to clarify the applicability and generalizability of these findings, particularly regarding their relevance in other farm settings or environments.
8 Lines 156-158: Please remove "Cycle threshold" and "Family Oral Fluids" from Figure 1. The previous mention of these terms included their full names, and now only the abbreviations are required.
9 Lines 190-192: It is necessary to include references to support the information provided.
10 Lines 198-199: It is necessary to include references to support the information provided.
11 Please review and correct the English grammar throughout the manuscript, paying attention to writing format issues.
Author Response
Dear reviewer,
Thank you very much for your comments and help to enhance the quality of our manuscript. Here are the answers to your different comments:
1) In the introduction, it would be appropriate for the author to provide an introduction to the epidemiological characteristics of PRRSV, including its significant impact on pig herd health and productivity, as well as its global prevalence. The author can also mention the two subtypes of PRRSV, namely PRRSV-1 and PRRSV-2, and elaborate on their differences and implications.
We added some information to answer your request L35-40.
2) In the "2.1. Study design" section of the Materials and Methods, please explain why this specific pig farm was chosen for the study, including background information on PRRSV-1 infection and instability at the site, as well as the reasons for not using PRRSV vaccines at the farm.
We added some information to answer your request L73-75.
3) In the "2.4. Pooling" section of the Materials and Methods, explain why the pooling ratios of 1:3 and 1:5 were chosen, and discuss the impact of these ratios on virus detection capability.
We chose these modalities because they are the most commonly used worldwide and to be able to compare our results with other researches with similar design (especially 1:5). We precised this on L225-227.
4) Line 153: The result indicates that the Ct values for the blood and FOF samples are 24 and 31, respectively. However, the significance of these values in PRRSV detection and an explanation for them are not provided. It would be helpful to have more information about Ct values to gain a better understanding of the result.
In this section, we mentioned the lowest Ct values for both sample types. We added a comment in the discussion part L209-213 regarding the significance of these data.
5) Lines 213-219: This section discusses the advantages and limitations of using FOF for PRRSV monitoring, as well as the effectiveness of using pooled samples. However, there is not enough explanation and discussion provided to support these conclusions, and no specific information or data is provided regarding these impacts.
We added some information as requested L 209-213.
6) In the conclusions, it is noted that the research results were obtained under specific conditions. However, there is a lack of sufficient information provided to clarify the applicability and generalizability of these findings, particularly regarding their relevance in other farm settings or environments.
We added a sentence L240-241.
7) Lines 156-158: Please remove "Cycle threshold" and "Family Oral Fluids" from Figure 1. The previous mention of these terms included their full names, and now only the abbreviations are required.
I am not sure that abbreviations are sufficient when reading a table or a figure. I let the editor inform us about this.
8) Lines 190-192: It is necessary to include references to support the information provided.
The Almeida reference is already cited in the text. Do you think it is not sufficient?
9) Lines 198-199: It is necessary to include references to support the information provided.
Reference is already cited, what do you mean exactly please?
10) Please review and correct the English grammar throughout the manuscript, paying attention to writing format issues.
Done, thank you
Reviewer 2 Report
This work, while not a novel idea, is important. Knowing PRRSV status for a pig producer is crucial. Many submitters choose to pool samples as stated in the manuscript, even sometimes being advised not to do so because of potential dilution effect and the risk of false negative results. This study, using a range of Cts for oral fluids and sera for pooling samples, drives home the risk of pooling for low titer samples.
I do have a few questions for clarification of the study:
line 104: What was the g-force used for FOF centrifugation?
lines 112-113: What were analysis settings for the PCR? Threshold, baseline?
line 115: How was negative status of the serum and oral fluid samples established? Was this by single PCR test for each sample or multiple tests? Samples of low titer don't always test positive by a single test, so duplicate or preferably triplicate testing would be preferred to be more confident in negative status.
Were samples ever subjected to freezing during the course of the study? Freeze/thaw cycles can effect PCR testing results/
Table 3: Is FOF=9 supposed to be Batch 4? The table has it as Batch 1, with no Batch 4 listed in the table.
The paper published by Gerber et. al (2013) should also be included in the references and discussion: Gerber PF, O'Neill K, Owolodun O, Wang C, Harmon K, Zhang J, Halbur PG, Zhou L, Meng XJ, Opriessnig T. Comparison of commercial real-time reverse transcription-PCR assays for reliable, early, and rapid detection of heterologous strains of porcine reproductive and respiratory syndrome virus in experimentally infected or noninfected boars by use of different sample types. J Clin Microbiol. 2013 Feb;51(2):547-56. doi: 10.1128/JCM.02685-12. Epub 2012 Dec 5. PMID: 23224085; PMCID: PMC3553906.
Very minor edits needed for this.
Author Response
Dear reviewer,
Thank you very much for your comments and help to enhance the quality of our manuscript. Here are the answers to your different comments:
I do have a few questions for clarification of the study:
line 104: What was the g-force used for FOF centrifugation?
We are not sure to understand your comment. G-force is already cited on L 111
lines 112-113: What were analysis settings for the PCR? Threshold, baseline?
These data are listed in manufacturer’s instructions procedure. Do you think necessary to add them in the text when using a commercial PCR test?
line 115: How was negative status of the serum and oral fluid samples established? Was this by single PCR test for each sample or multiple tests? Samples of low titer don't always test positive by a single test, so duplicate or preferably triplicate testing would be preferred to be more confident in negative status.
You are right, thank you for your comment. Negative status was established by a single PCR test, as regularly done in the field.
Were samples ever subjected to freezing during the course of the study? Freeze/thaw cycles can effect PCR testing results/
Only cool storage. The information was noticed in the manuscript for FOF and UW but not for blood samples. We added it on L87-88.
Table 3: Is FOF=9 supposed to be Batch 4? The table has it as Batch 1, with no Batch 4 listed in the table.
Thank you for your attentive review. We have made the correction.
The paper published by Gerber et. al (2013) should also be included in the references and discussion: Gerber PF, O'Neill K, Owolodun O, Wang C, Harmon K, Zhang J, Halbur PG, Zhou L, Meng XJ, Opriessnig T. Comparison of commercial real-time reverse transcription-PCR assays for reliable, early, and rapid detection of heterologous strains of porcine reproductive and respiratory syndrome virus in experimentally infected or noninfected boars by use of different sample types. J Clin Microbiol. 2013 Feb;51(2):547-56. doi: 10.1128/JCM.02685-12. Epub 2012 Dec 5. PMID: 23224085; PMCID: PMC3553906
Thank you for your suggestion. We added this reference.
Reviewer 3 Report
PRRSV antigen detection has been a challenge for the global pig industry. In this paper, the use of FOF and UW instead of serum sample detection can avoid exogenous infection during blood collection and save labor costs, which is of certain significance.
Major revision:
The biggest problem with using FOF to replace serum is poor sensitivity. However, only 13 out of 120 serum samples were positive. The results in lines 145-148 of this article are meaningless. If the serum test results are used as the gold standard, the positive coincidence rate of the two tests is only 46%. The authors need to collect a large number of positive samples for testing to prove that FOF is a good substitute for serum.
At the same time, it is suggested that the authors choose animal experimental samples with known background for testing, and then test clinical samples after proving the substitution of FOF for serum.
Minor revision:
1. FOF, UW, PF, UC and TT sample types were mentioned in lines 44-46 of the paper. In lines 56-58, it was also mentioned that few studies used OF, TT and UC sample types to detect PRRSV-1. In this paper, serum, FOF and UW were tested. May I ask Why isn 't PF tested together when designing experiments?
2. In Table 3, the Ct values of serum 7,8, and 9 in undiluted condition were all 30, and the Ct values of serum 7, 8 in 1:3 and 1:5 dilution were similar and positive, but the Ct values of serum 9 in 1:3 and 1:5 dilution were both greater than 40, which was considered negative. In addition, the Ct value of serum 10 was 29 when it was undiluted, but no Ct value was detected after dilution. Has the abnormal result been reexamined? If so, why does this situation occur? If the result is due to improper storage of nucleic acid samples or other human factors, then the overall test results in Table 3 lack confidence?
Author Response
Dear reviewer,
Thank you very much for your comments and help to enhance the quality of our manuscript. Here are the answers to your different comments:
Major revision:
The biggest problem with using FOF to replace serum is poor sensitivity. However, only 13 out of 120 serum samples were positive. The results in lines 145-148 of this article are meaningless. If the serum test results are used as the gold standard, the positive coincidence rate of the two tests is only 46%. The authors need to collect a large number of positive samples for testing to prove that FOF is a good substitute for serum.
At the same time, it is suggested that the authors choose animal experimental samples with known background for testing, and then test clinical samples after proving the substitution of FOF for serum.
We did not aim to prove that FOF is a good alternative even if it is widely used in some countries. On the contrary, we try to compare the detection rates at different levels (litter, batch) and moderate the interest of FOF, at least in our tested herd.
Minor revision:
- FOF, UW, PF, UC and TT sample types were mentioned in lines 44-46 of the paper. In lines 56-58, it was also mentioned that few studies used OF, TT and UC sample types to detect PRRSV-1. In this paper, serum, FOF and UW were tested. May I ask Why isn 't PF tested together when designing experiments?
Just because castration is less and less common in our country regarding welfare considerations.
- In Table 3, the Ct values of serum 7,8, and 9 in undiluted condition were all 30, and the Ct values of serum 7, 8 in 1:3 and 1:5 dilution were similar and positive, but the Ct values of serum 9 in 1:3 and 1:5 dilution were both greater than 40, which was considered negative. In addition, the Ct value of serum 10 was 29 when it was undiluted, but no Ct value was detected after dilution. Has the abnormal result been reexamined? If so, why does this situation occur? If the result is due to improper storage of nucleic acid samples or other human factors, then the overall test results in Table 3 lack confidence?
I understand very well your concern. Nonetheless, be cautious because ND = not done for serum 10 because of insufficient quantity of serum (it was already precised in L168). Finally, regarding comparative Ct value of serum 9 before and after pooling, for sure it is questionable but the result is the result and I am not sure that this fact makes all the results unconfident. We retested the pools for this serum two times with the same results.
Thank you again
Round 2
Reviewer 1 Report
good
Reviewer 3 Report
你立即认真而真诚地修改了。因此,现在是接受在兽医科学上发表的时候了。